**Data Availability Statement:** The data underlying the findings in our study are not publicly available because the original approval by the regional

# Latent profile analysis patterns of exercise, sitting and fitness in adults – Associations with metabolic risk factors, perceived health, and perceived symptoms

Elin Ekblom-Bak[1]⊙*, Andreas Stenling[2]⊙, Jane Salier Eriksson[1]‡, Erik Hemmingsson[1]‡, Lena V. Kallings[1]‡, Gunnar Andersson[3]‡, Peter Wallin[3]‡, Örjan Ekblom[1]‡, Björn Ekblom[1]‡, Magnus Lindwall[1,4]⊙

**1** Åstrand Laboratory of Work Physiology, The Swedish School of Sport and Health Sciences, Stockholm, Sweden, **2** Department of Psychology, Umeå University, Umeå, Sweden, **3** Research Department, HPI Health Profile Institute, Danderyd, Sweden, **4** Department of Psychology, University of Gothenburg, Gothenburg, Sweden

⊙ These authors contributed equally to this work.
‡ These authors also contributed equally to this work.
* eline@gih.se

## Abstract

### Aim

To identify and describe the characteristics of naturally occurring patterns of exercise, sitting in leisure time and at work and cardiorespiratory fitness, and the association of such profiles with metabolic risk factors, perceived health, and perceived symptoms.

### Methods

64,970 participants (42% women, 18–75 years) participating in an occupational health service screening in 2014–2018 were included. Exercise and sitting were self-reported. Cardiorespiratory fitness was estimated using a submaximal cycle test. Latent profile analysis was used to identify profiles. BMI and blood pressure were assessed through physical examination. Perceived back/neck pain, overall stress, global health, and sleeping problems were self-reported.

### Results

Six profiles based on exercise, sitting in leisure time and at work and cardiorespiratory fitness were identified and labelled; Profile 1 "*Inactive, low fit and average sitting in leisure, with less sitting at work*"; Profile 2 "*Inactive, low fit and sedentary*"; Profile 3 "*Active and average fit, with less sitting at work*"; Profile 4 "*Active, average fit and sedentary in leisure, with a sedentary work*" (the most common profile, 35% of the population); Profile 5 "*Active and fit, with a sedentary work*"; Profile 6 "*Active and fit, with less sitting at work*". Some pairwise similarities were found between profiles (1 and 2, 3 and 4, 5 and 6), mainly based on similar levels of exercise, leisure time sitting and fitness, which translated into similar dose-response

ethic's board (Stockholm Ethics Review Board, Dnr 2015/1864-31/2 and 2016/9-32) and the informed consent from the subjects participating in the studies did not include such a direct, free access. If a reader wants access to the data underlying the present article, please contact the HPI Health Profile Institute at support@hpihealth.se.

**Funding:** This work was supported by The Swedish Research Council for Health, Working Life and Welfare https://forte.se/en/ (Grant no 2018-00384, grant recieved by author EEB) and The Swedish Heart-Lung Foundation https://www.hjart-lungfonden.se/HLF/Om-Hjart-lungfonden/About-HLF/ (Grant no 20180636, grant recieved by author EEB). AS is supported by an international postdoc grant from the Swedish Research Council www.vr.se/english.html (Grant no2017-00273). The funders had no role in study design, data collection and analysis, decision to publish, or preparation of the manuscript. The funder HPI Health Profile Institute provided support in the form of salaries for authors [GA, PW] and research materials, but did not have any additional role in the study design, data collection and analysis, decision to publish, or preparation of the manuscript. The specific roles of these authors are articulated in the 'author contributions' section.

**Competing interests:** I have read the journal's policy and the authors of this manuscript have the following competing interests: Gunnar Andersson (responsible for research and method) and Peter Wallin (CEO and responsible for research and method) are employed at HPI Health Profile Institute. This does not alter our adherence to PLOS ONE policies on sharing data and materials.

associations with the outcomes. In general, profile 1 and 2 demonstrated most adverse metabolic and perceived health, profile 4 had a more beneficial health than profile 3, as did profile 6 compared to profile 5.

## Conclusions

The present results implies a large variation in exercise, sitting, and fitness when studying naturally occurring patterns, and emphasize the possibility to target exercise, sitting time, and/or fitness in health enhancing promotion intervention and strategies.

## Introduction

Exercise, sedentary time and cardiorespiratory fitness are independent predictors of metabolic health and cardiovascular disease risk [1–3]. Recent trend analyses indicate increased time spent sedentary, both at work and in leisure time [4, 5], sustained or decreased levels of moderate-to-vigorous physical activity [4, 6], and decreased levels in cardiorespiratory fitness [7] over the last decade, which may have induced a greater variability in daily physical activity patterns and physical performance between individuals. As these physical activity-related variables tend to cluster, analyses on the interaction and the interactive effect of various physical activity variables on health and disease risk are highly relevant. Traditionally, this has been investigated through variable-centered analyses where interaction effects typically were tested in regression models or stratified analyses. Although this strategy may provide valuable information, testing, and interpreting interactions consisting of more than two variables is challenging. Person-oriented approaches offer an alternative way to understand how different variables interact that may complement and extend variable-centered research [8, 9]. Although cluster-analysis has traditionally been the most commonly used choice of method in person-centered research, other types of analyses, such as latent profile analysis (LPA), are today generally recommended as they entail a number of benefits compared to cluster analyses [10, 11].

LPA identifies interactions of input variables (e.g., exercise, fitness, and sedentariness) to create naturally occurring profiles, or typical patterns, of combinations of different variables, in a heterogeneous population [10, 11]. This is particularly useful for complex within-person interactions, including more than two variables at the time. Previous studies in adults have mainly applied LPA to identify how physical activity and sedentary behaviour varies in different patterns over 7 days of the week, and how combinations of physical activity and sedentary behaviour are associated with metabolic health and longevity [12–14]. A few studies have used LPA to identify patterns of variations in physical activity, commuting, energy expenditure, and sedentary time [15, 16]. However, no previous study have used LPA to study the naturally occurring patterns of sedentary time at work and during leisure time, exercise (defined as physical activity that is planned, structured, repetitive, and that favours physical fitness maintenance), and cardiorespiratory fitness. Also, the way such patterns relate to health are not known. Such knowledge would expand the understanding how different physical activity-related variables naturally interacts and associate with health, which in turn could be used to target specific high-risk populations and for future intervention strategies.

Hence, in a large population of men and women of different ages, we aimed to identify and describe the characteristics of naturally occurring patterns/latent profiles of exercise, sitting in leisure time and at work, and fitness ($VO_2max$). Also, we aim to study the association of such profiles with metabolic risk factors, perceived health, and perceived symptoms.

## Material and methods

This study has a cross-sectional study design. Data was derived from the HPI Health Profile Institute database, which comprises data from Health Profile Assessments (HPAs) carried out in Swedish health services all over Sweden. The test protocol, methods used and education of HPA coaches is standardized and has been the same since the start of HPA in the middle of the 1970s. The participant answers an extensive questionnaire including current lifestyle, physical activity habits, and perceived symptoms and health, followed by a dialogue with a HPA coach, who also collected data on height, body weight, and diastolic and systolic blood pressure (see below). Cardiorespiratory fitness was estimated using submaximal testing. After completion of the HPA, all data were subsequently recorded in the HPI database. Participation in HPA was offered to all employees working for a company or an organisation connected to occupational or other health services in Sweden, and free of charge for the employee.

In January 2014, questions regarding self-reported sitting time at work and during leisure time were added to the protocol. In April 2018, a withdrawal of all participants with one HPA registered in the HPI database since January 2014 and with valid data for the self-reported sitting questions was made for the present analyses ($N$ = 84,937). Of these, n = 188 lacked data on self-reported exercise, n = 19,526 on valid fitness estimate, and n = 253 of the descriptive (smoking and diet) or outcome (BMI, systolic and diastolic blood pressure, symptom back/neck, overall stress, and perceived health) variables, resulting in n = 64,970 participants (42% women, aged 18–75 years) included in the latent profile analysis. The protocols used were approved by the institutional review boards of the institutions involved in this study and all participants provided informed consent prior to data collection. The study was approved by the Stockholm Ethics Review Board (Dnr 2015/1864-31/2 and 2016/9-32), and adhered to the Declaration of Helsinki.

### Latent profile analysis input variables

Current exercise habits were self-reported through the statement; *I exercise for the purpose of maintaining/improving my physical fitness, health and well-being . . .* with the alternatives 1 = *Never*, 2 = *Sometimes*, 3 = *1–2 times/week*, 4 = *3–5 times/week* or 5 = *≥6 times/week*. Workplace and leisure time sitting were self-reported through the statements: *I sit at work. . .* and *I sit in my leisure time. . .*, with the alternatives 1 = *Almost all of the time*, 2 = *75% of the time*, 3 = *50% of the time*, 4 = *25% of the time* or 5 = *Almost none of the time*. The exercise question has yet not been validated against objective measures. The sitting questions were derived from the question previously showed to have high predictive validity by Katzmarzyk et al. [17], and strong convergent validity with total and prolonged stationary time [18]. Cardiorespiratory fitness (hereby referred to as fitness) was estimated using the Åstrand submaximal cycle ergometer test [19]. All participants were requested to refrain from vigorous activity the day before the test, heavy meal three hours and smoking/snuff use one hour before the test, and avoiding stressing to the test. The participant cycled on a calibrated ergometer at an individually, chosen submaximal work rate for six minutes to achieve a steady-state pulse. Using the steady-state pulse, $VO_2max$ was estimated from a sex-specific nomogram, with corresponding age-correction factors, expressed as absolute (L·min$^{-1}$) and relative (ml·min$^{-1}$·kg$^{-1}$) $VO_2max$. Criterion validity has been tested for the Åstrand test, showing no systematic bias and limited variation in mean difference between estimated and directly measured $VO_2max$, mean difference 0.01 L $O_2$ min$^{-1}$ (95% CI -0.10 to 0.11) [20, 21].

### Anthropometrics and blood pressure

Body mass was assessed with a calibrated scale in light-weight clothing to the nearest 0.5 kg, and body height measured to the nearest 0.5 cm using a wall-mounted stadiometer. BMI

(kg·m$^{-2}$) was subsequently calculated. Waist circumference was measured to the nearest 0.5 cm with a tape measure at the midpoint between the top of the iliac crest and the lower margin of the last palpable rib in the mid axillary line after normal exhalation. Systolic and diastolic BP (mmHg) was measured manually in the right arm using the standard auscultatory method after 20 minutes of seated resting.

## Perceived health and symptoms

Perceived global health was assessed through the statement: *I perceive my physical and mental health as . . .* with the alternatives *Very poor*, *Poor*, *Neither good or bad*, *Good*, or *Very good*. Perceived back/neck pain, overall stress and sleeping problems were obtained through the statements: *I have back/neck issues. . .*, *I perceive stress in my life, both personally and at work. . .* and *I perceive sleeping difficulties. . .*, with the alternatives *Very often*, *Often*, *Sometimes*, *Rarely* or *Never*.

## Other variables

Physical activity level prior to the age of 20 years was self-reported by selecting one of the following five given alternatives through the statement; *Prior to the age of 20, I. . . Did not participate in physical education class*, *Participate only in physical education class*, *Participate in physical education class + 1–2 times/week of physical activity outside school hours*, *Participate in physical education class + 3–5 times/week of physical activity outside school hours* or *Participate in physical education class + At least 6 times/week of physical activity outside school hours*. This question has previously shown predictive validity for exercise level, fitness and health later in life [22]. Dietary habits were obtained using the statement *I consider my diet, regarding both meal frequency and nutritional content to be . . .* with the alternatives *Very poor*, *Poor*, *Neither good nor bad*, *Good* or *Very good*. Smoking habits were obtained using the statement *I smoke . . .* with the alternatives *At least 20 cig/day*, *11–19 cig/day*, *1–10 cig/day*, *Occasionally* or *Never*.

In addition, data were also obtained from national quality control registers, including occupation, which was reported according to the Swedish Standard Classification of Occupations 1996 (SSYK96) until June 2014 and according to the SSYK 2012 after that. SSYK is a system for classifying and aggregating data about occupations in administrative registers or statistical surveys. Occupations reported according to both SSYK96 or SSYK 2012 can be further grouped into four broad skill levels defined by level of education for the particular occupation; *Level 1* covers elementary education at primary school level, meaning no or a low formal education requirements, *Level 2* covers education programs at upper secondary and tertiary level of no more than 2 years in length, *Level 3* covers practical or vocational tertiary education programs of 2–3 years in length, and *Level 4* covers theoretical or research-oriented tertiary education programs and third-cycle programs of at least 3 years, normally 4 years or longer in length.

## Statistical analysis

Mplus version 8.3 was used to estimate the LPA and group individuals into profiles based on their individual patterns of exercise, sitting time, and fitness [23]. A sequence of nested models with an increasing number of profiles were compared to examine if more complex models (i.e., with more profiles) fit the data better than more parsimonious models (i.e., with less profiles). In the present study, models with 1 to 7 latent profiles were tested to identify the optimal number of profiles. Important criteria for determining the number of latent profiles in the data were substantive meaning and theoretical conformity [10] as well as statistical adequacy of the solution (e.g., absence of negative variance estimates and local solutions; e.g., [24–26]).

For decisions about model retention we followed recommendations [27, 28] and relied on several indexes: the consistent Akaike Information Criterion, the Bayesian Information Criterion, and the sample-size adjusted Bayesian Information Criterion. Lower values for these indexes indicate better model fit. The entropy criterion was also examined, which indicates the precision of the classification of cases into the different profiles. However, the entropy should not be used when determining the optimal number of profiles [28, 29], but can be a useful tool to assess classification accuracy. Higher entropy values indicate fewer classification errors. When merging the total information regarding the most likely number of profiles in the data, subjective dimensions of choice beyond fit statistics were necessarily involved. More parsimonious models with fewer profiles were chosen over more complex profiles where this enhanced the interpretability of the profiles.

To support the interpretation of the best-fitting solution, *z*-scores of the observed variables were used. We are not aware of any agreed upon criteria for low and high values, but we interpreted the scores of the input variables in relative terms, compared to the group means. Following trends in previous studies using LPA [30], standardized scores between +0.5 and -0.5 were labelled as average, scores over 0.5 were labelled as (relatively) high, and scores lower than -0.5 were labelled as (relatively) low. The profiles retained in the final solution were characterized by sex, age, SSYK level, smoking, diet, and activity prior to 20 years of age (Table 1), and metabolic risk factors, perceived health, and perceived symptoms (Table 2). Statistically significant differences between the profiles for continuous data were tested for by general linear modelling (parametric) and Kruskal-Wallis ANOVA (non-parametric) with pairwise comparison adjusting for multiple comparisons, and for proportions by comparing proportions with the 99% confidence interval (CI) to compensate for multiple testing. Logistic regression modelling was used to assess the odds ratio (OR) and 95% CI associated with the different profiles (using profile 2 as reference) for dichotomized variables of BMI ($>$ 30 vs. $\leq$ 30 kg·m$^{-2}$), high systolic ($\geq$140 vs. $<$140 mmHg) or diastolic ($\geq$90 vs. $<$90 mmHg) blood pressure, global health (Very poor/Poor vs. Neither good or bad/Good/Very good), back/neck pain (Very often/Often vs. Sometimes/Rarely/Never), sleeping problems (Very often/Often vs. Sometimes/Rarely/Never), and overall stress (Very often/Often vs. Sometimes/Rarely/Never). The ORs were adjusted for sex and age, and further adjusted for diet, smoking, and educational level (multi-adjusted). Comparisons between profiles were performed using IBM SPSS (version 24.0, SPSS Inc., Chicago IL) and Confidence Interval Analysis (version 2.2.0).

## Results

The information criteria decreased for each additional profile, indicating a better model fit with more profiles (S1 Table). However, the model with seven profiles converged on a local solution (i.e., the best log likelihood value was not replicated despite that multiple start values were considered) rather than a global solution, which likely is an indication that too many profiles are being extracted [26, 31]. Hence, based on the statistical adequacy criteria [24], we did not estimate more than six profiles. We also compared the six-profile solution to the five-profile solution to examine the interpretability and meaningfulness of adding a sixth profile. Adding a sixth profile provided a theoretically interpretable and meaningful additional profile that differed not only in level but also in shape when compared to the profiles in the five-profile solution. Hence, based on the combined information from the statistical criteria and interpretability we retained the six-profile solution as our final model. The six profiles are presented in Fig 1. Positive z-scores indicates a more beneficial behaviour/level and relatively higher scoring (more exercise, less sitting in leisure and at work, and higher fitness), and negative z-scores indicates a less beneficial behaviour/level and relatively lower scoring (less exercise, more

**Table 1. Characteristics of the different profiles (above) and mean with standard errors of the z-score for the latent profile variables (below).**

| | Profile 1 (n = 9480) | Profile 2 (n = 11227) | Profile 3 (n = 14182) | Profile 4 (n = 23017) | Profile 5 (n = 4871) | Profile 6 (n = 2193) |
|---|---|---|---|---|---|---|
| Women | 33% (3137)[a,b,c,d,e] | 39% (4400)[b,c,d] | 43% (6113)[c,d,e] | 45% (10438)[e] | 45% (2191)[e] | 37% (816) |
| Age (years) | 43.2 (0.12)[a,b,c,d,e] | 44.2 (0.10)[d,e] | 44.1 (0.10)[d,e] | 44.4 (0.07)[d,e] | 37.0 (0.13)[e] | 34.6 (0.22) |
| SSYK 1–3 | 20% (1447/7337) | 63% (4667/7382) | 29% (3145/10762) | 72% (10632/14842) | 78% (2423/3112) | 33% (542/1659) |
| Non-smokers | 75% (7071)[a,b,c,d,e] | 83% (9311)[b,c,d] | 85% (12010)[c,d] | 89% (20574)[d,e] | 91% (4430)[e] | 84% (1851) |
| Very good/good diet | 57% (5363)[b,c,d,e] | 58% (6493)[b,c,d,e] | 75% (10641)[d,e] | 74% (17061)[d,e] | 84% (4069) | 82% (1786) |
| Leisure-time physically active in youth | 79% (3355/4227)[a,b,c,d,e] | 82% (4016/4920)[b,c,d,e] | 86% (5766/6738)[c,d,e] | 87% (9003/10299)[d,e] | 93% (1971/2115) | 92% (926/1010) |
| **Profile variables** | | | | | | |
| Exercise (1 = Never to 5 = ≥6 times/week) | 2 (1–2)[b,c,d,e] | 2 (1–2)[b,c,d,e] | 4 (3–4)[c,d,e] | 3 (3–4)[d,e] | 4 (3–4) | 4 (4–4) |
| Cardiorespiratory fitness (ml·min$^{-1}$·kg$^{-1}$) | 31.4 (0.08)[b,c,d,e] | 31.3 (0.07)[b,c,d,e] | 34.2 (0.06)[c,d,e] | 35.1 (0.05)[d,e] | 54.1 (0.08) | 54.5 (0.13) |
| Leisure sitting (1 = All time to 5 = No time) | 4 (3–4)[a,b,d,e] | 3 (3–4)[b,c,d,e] | 4 (3–4)[c,d,e] | 4 (3–4)[d,e] | 4 (3–4)[e] | 4 (4–5) |
| Workplace sitting (1 = All time to 5 = No time) | 5 (4–5)[a,b,c,d] | 2 (2–3)[b,e] | 5 (4–5)[c,d] | 2 (2–3)[d,e] | 2 (1–3)[e] | 5 (4–5) |
| **Profile variables in z-score** | | | | | | |
| Exercise | -1.086 (0.010) | -1.064 (0.011) | 0.603 (0.009) | 0.424 (0.008) | 0.740 (0.016) | 0.819 (0.021) |
| Cardiorespiratory fitness | -0.473 (0.009) | -0.495 (0.011) | -0.135 (0.014) | -0.025 (0.013) | 1.519 (0.032) | 1.547 (0.044) |
| Leisure sitting | -0.108 (0.012) | -0.472 (0.015) | 0.278 (0.009) | -0.037 (0.010) | 0.272 (0.017) | 0.516 (0.018) |
| Workplace sitting | 0.984 (0.007) | -0.778 (0.009) | 0.928 (0.007) | -0.721 (0.007) | -0.778 (0.014) | 0.931 (0.015) |

Data is presented as mean (SE) (parametric), median (Q1-Q3) (non-parametric) and % (n). For variables with missing data, % (n/N) is presented.

[a] significant different vs. Profile 2 after Bonferroni correction for multiple comparisons.

[b] significant different vs. Profile 3 after Bonferroni correction for multiple comparisons.

[c] significant different vs. Profile 4 after Bonferroni correction for multiple comparisons.

[d] significant different vs Profile 5 after Bonferroni correction for multiple comparisons.

[e] significant different vs Profile 6 after Bonferroni correction for multiple comparisons.

sitting in leisure and at work, and lower fitness,), compared to average. Z-scores around zero means average scoring/level.

## The profiles and their characteristics

Profile 1 and 2 share similar patterns with low relative levels of exercise (median 2 ="Sometimes" for both profiles), low relative fitness (mean 31.4 and 31.3 ml·min$^{-1}$·kg$^{-1}$, respectively) and average/high relative levels of leisure time sitting (median 4 ="25% of the time" for profile 1 and 3 ="50% of the time"), but differs regarding sitting at work (median 5 ="Almost none of the time" for profile 1 and 2 ="75% of the time" for profile 2). Consequently, profile 1 may be labelled "*Inactive, low fit and average sitting in leisure, with less sitting at work*" and profile 2 "*Inactive, low fit and sedentary*". Moreover, while profile 3 had high relative levels of exercise (4 ="3–5 times/week" for profile 3), profile 4 had average levels of exercise (3 ="1–2 times/week" for profile 4), but with similar moderate relative levels of fitness (mean 34.2 and 35.1 ml·min$^{-1}$·kg$^{-1}$, respectively). The profiles had similar low relative levels of leisure time sitting (4 ="25% of the time" for both profiles), but with low relative levels of sitting at work for profile 3 (median 5 ="Almost none of the time") and high relative levels of sitting at work for profile 4 (median 2 ="75% of the time"). Consequently, profile 3 may be labelled "*Active and average fit, with less sitting at work*" and profile 4 "*Active, average fit and sedentary in leisure, with a sedentary work*". Profiles 5 and 6 shared similar patterns with high relative levels of exercise (median

**Table 2. Anthropometrics and blood pressure (above), and perceived symptoms and global health (below) in relation to the six profiles.**

| | Profile 1 | Profile 2 | Profile 3 | Profile 4 | Profile 5 | Profile 6 |
|---|---|---|---|---|---|---|
| | (n = 9480) | (n = 11227) | (n = 14182) | (n = 23017) | (n = 4871) | (n = 2193) |
| Body Mass Index (kg·m$^{-2}$) | 26.9 (0.04)[b,c,d,e] | 26.9 (0.04)[b,c,d,e] | 26.2 (0.03)[c,d,e] | 25.8 (0.03)[d,e] | 23.4 (0.06)[e] | 23.7 (0.09) |
| BMI>30 | 22% | 22% | 16% | 13% | 1% | 1% |
| Diastolic BP (mmHg) | 79.7 (0.10)[a,b,c,d,e] | 79.2 (0.09)[b,c,d,e] | 78.4 (0.08)[c,d,e] | 78.0 (0.06)[d,e] | 74.8 (0.14)[e] | 75.5 (0.21) |
| Systolic BP (mmHg) | 126.9 (0.14)[a,b,c,d,e] | 125.5 (0.13)[b,d,e] | 126.1 (0.10)[c,d,e] | 125.0 (0.09)[d,e] | 121.7 (0.73)[e] | 123.5 (0.29) |
| High diastolic or systolic BP[#] | 28% | 25% | 23% | 21% | 7% | 9% |
| Symptoms, 1 = Very often to 5 = Never | | | | | | |
| Symptom Back/Neck | 3.27 (0.01)[a,b,c,d,e] | 3.33 (0.01)[b,c,d,e] | 3.45 (0.01)[c,d,e] | 3.51 (0.01)[d,e] | 3.64 (0.02) | 3.64 (0.02) |
| Very often/often | 23% | 23% | 20% | 18% | 14% | 13% |
| Sleeping problems | 3.95 (0.01)[b,d,e] | 3.91 (0.01)[b,c,d,e] | 4.01 (0.01)[e] | 3.98 (0.01)[d,e] | 4.05 (0.02) | 4.12 (0.03) |
| Very often/often | 11% | 11% | 9% | 9% | 6% | 6% |
| Overall stress | 3.51 (0.01)[a,b,e] | 3.34 (0.01)[b,c,d,e] | 3.62 (0.01)[c,d] | 3.48 (0.01)[d,e] | 3.44 (0.01)[e] | 3.63 (0.02) |
| Very often/often | 14% | 18% | 10% | 12% | 14% | 10% |
| Global health, 1 = Very poor to 5 = Very good | | | | | | |
| Overall health | 3.52 (0.01)[a,b,c,d,e] | 3.44 (0.01)[b,c,d,e] | 3.87 (0.01)[c,d,e] | 3.83 (0.01)[d,e] | 4.08 (0.01)[e] | 4.15 (0.02) |
| Very poor/poor | 10% | 12% | 4% | 4% | 2% | 1% |

All data is presented as mean (SE), and adjusted for sex and age (GLM).

BP, blood pressure.

[a] significant different vs. Profile 2 after Bonferroni correction for multiple comparisons.

[b] significant different vs. Profile 3 after Bonferroni correction for multiple comparisons.

[c] significant different vs. Profile 4 after Bonferroni correction for multiple comparisons.

[d] significant different vs Profile 5 after Bonferroni correction for multiple comparisons.

[e] significant different vs Profile 6 after Bonferroni correction for multiple comparisons.

[#] Diastolic BP ≥90 mmHg and/or Systolic BP ≥140 mmHg.

4 = "3–5 times/week" for both profiles), high relative fitness (mean 54.1 and 54.5 ml·min$^{-1}$·kg$^{-1}$, respectively) and low relative levels of leisure time sitting (4 = "25% of the time" for both profiles), but differs in terms of sitting at work (median 2 = "75% of the time" for profile 5 and 5 = "Almost none of the time" for profile 6). Consequently, profile 5 may be labelled "*Active and fit, with a sedentary work*" and profile 6 "*Active and fit, with less sitting at work*".

The characteristics of the different profiles is presented in Table 1.

## Associations with metabolic risk factors, perceived health and perceived symptoms

The six profiles were further associated to continuous (Table 2) and dichotomized (Fig 2) levels of metabolic risk factors, as well as perceived health and symptoms. For the continuous variables, Profile 1 and 2 demonstrated more adverse health, with specifically higher prevalence of obesity and very poor/poor perceived overall health, compared to the other profiles. Profile 4 had in general a more beneficial health and symptom profile than profile 3. Profile 5 had a more beneficial metabolic profile, but scored more overall stress and poorer overall health, compared to profile 6.

In Fig 2, profile 2 was set as reference because it had the least favourable profile based on the four input variables. In general, profile 1 and 2 had the highest OR for all outcomes, compared to the other profiles in multi-adjusted analyses. Profile 2 had highest OR for obesity, overall stress, and poor global health, while profile 1 had highest OR for high blood pressure. No

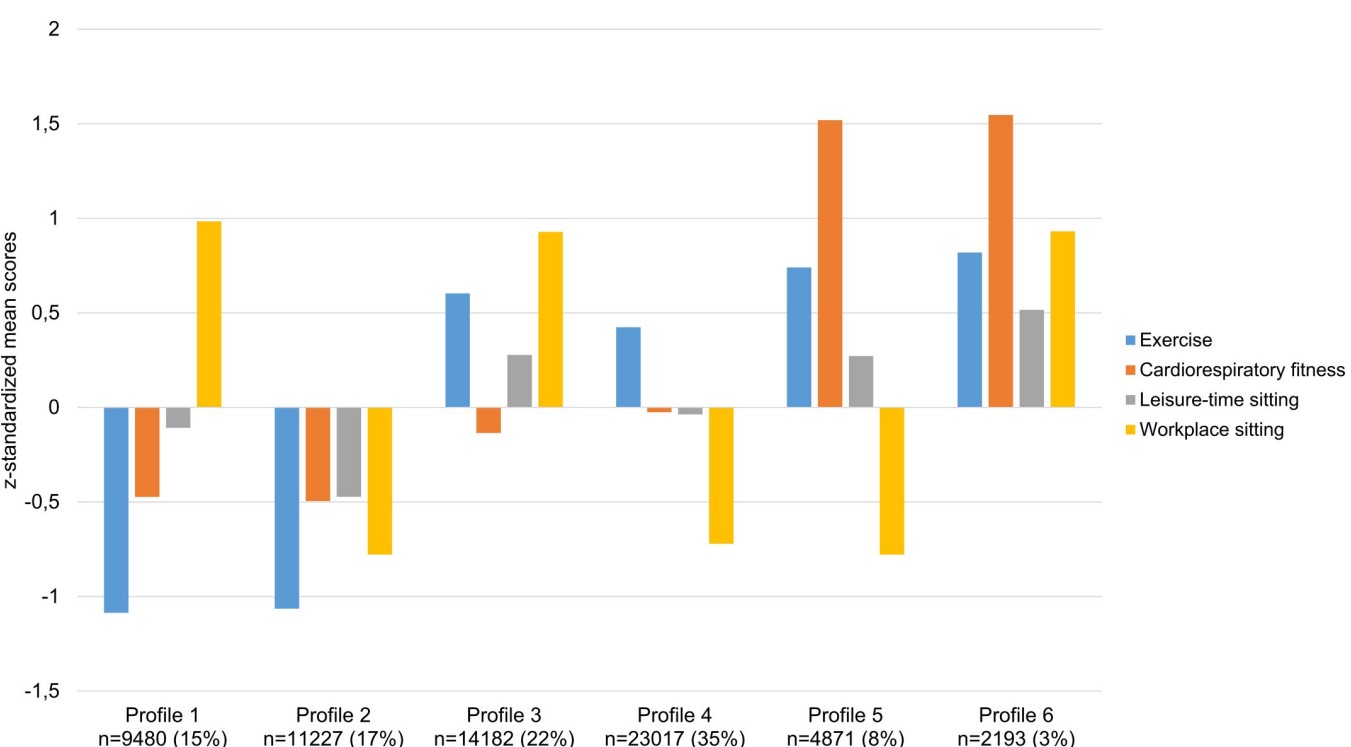

**Fig 1. Illustration of z-score distribution of self-reported exercise, sitting in leisure time, sitting at work, and cardiorespiratory fitness, in the six profiles defined in the latent profile analysis.** Percentages within parenthesis is in relation to the total sample.

differences were seen for back/neck pain and sleeping problems. Profile 3 and 4 had lower ORs for all outcomes compared to profile 1 and 2, with small differences between the two profiles. Further, profile 5 and 6 had lower ORs compared to profile 3 and 4 for obesity, high blood pressure, back/neck pain, and poor global health, but with similar OR as profile 3 and 4 for overall stress and sleeping problems. Differences between profile 5 and 6 were generally small.

A total of 19,892 participants lacked data on educational level and were hence excluded in the multi-adjusted analyses. A drop-out analysis was performed comparing core variables (including sex, age, profile, BMI, blood pressure, diet habits, smoking, and perceived stress, back/neck pain, global health, and sleeping problem) for participants included and excluded in the multi-adjusted analyses revealed significant but only marginal differences between the two groups (S2 Table).

## Discussion

In a large sample including more than 64,000 men and women, we used LPA and identified six distinct profiles based on exercise, sitting in leisure time and at work, and cardiorespiratory fitness. The number of profiles identified, and the variation of the input variables between the profiles, confirm the large variation in the interaction of exercise, sitting, and fitness between individuals. However some similarities were found; profile 1 and 2, profile 3 and 4 and profile 5 and 6, respectively, shared similar patterns regarding exercise, fitness, and leisure time sitting in a dose-response manner (from least to most beneficial), but differed regarding workplace sitting. This shared pattern translated into similar dose-response associations with OR especially for obesity, high blood pressure, neck/back pain, and poor perceived global health. Regarding often perceived overall stress and sleeping problems, profile 3 to 6 had similar lower OR

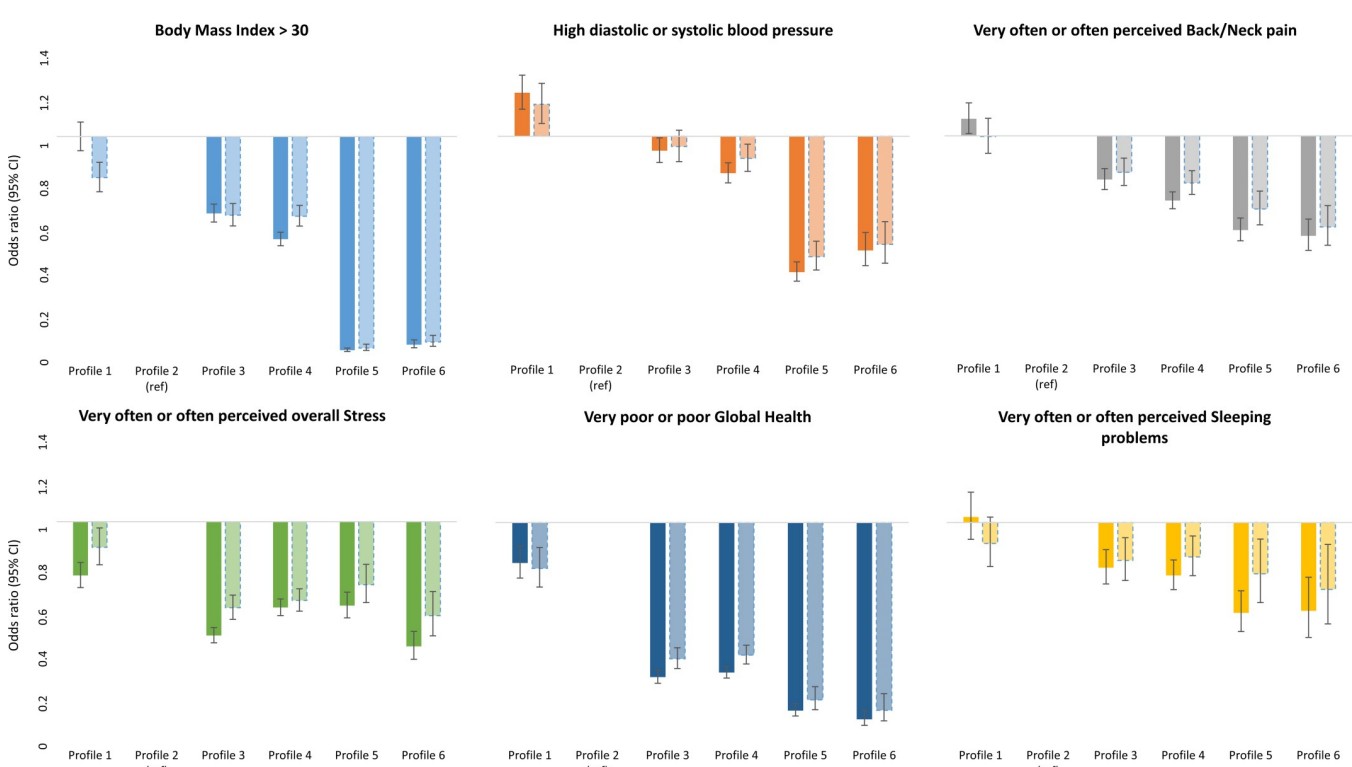

**Fig 2. Sex- and age adjusted (dark bars) and multi-adjusted (lighter bars) OR (95% CI) for six dichotomized risk factors.** The multi-adjusted ORs are further adjusted for diet, smoking and educational level.

compared to profile 2 (reference) and profile 1. These results emphasize the possibility to target exercise, sitting time, and/or fitness in health enhancing promotion intervention and strategies.

The difference in workplace sitting was only associated with the metabolic risk factors, perceived health, and perceived symptoms for profile 1 and 2 (which both had low exercise and fitness level). The less sitting at work for profile 1 transduced into lower OR for obesity, overall stress, and poor global health. This is somewhat contradictory to the recently proposed physical activity paradox which suggests beneficial health outcomes associated with high level leisure time physical activity, but detrimental health consequences with those exposed to high level occupational physical activity [32] and lower mortality risk in those with highest sitting time at work [33]. Though, other results suggests an increase in obesity and type 2 diabetes with higher sitting time at work [34]. However, further comparisons of profile 3 vs. 4, and profile 5 vs. 6, which had similar differences in sitting at work between the profiles, show no similar discrimination of the health and symptom outcomes by level of sitting at work as when comparing profile 1 and 2. This is probably due to the higher relative levels of exercise, fitness and lower relative levels of leisure time sitting compared to profile 1 and 2 and may mirror how different characteristics may buffer lack in another.

Previous studies have used LPA to study day-to-day pattern of physical activity and sedentary time over the week, using one type of domain/measure. For example, Metzger et al. concluded that 78% of their U.S. population-based sample was classified into patterns of accelerometer-derived physical activity that average less than 25 min per day of moderate-to-vigorous physical activity, which is considered low compared to guidelines [14]. Accumulating the recommended amount of weekly physical activity was consistently associated with a more beneficial metabolic health profile [35]. However, the manner in which the physical activity

was accumulated (spread over the week or in just a few days) had similar associations with metabolic risk. Similar results were reported for all-cause mortality risk, where time spent in more active classes (higher average physical activity intensity or percentages of time in moderate-to-vigorous intensity physical activity) reduced the risk and more time spent in sedentary bouts increased the risk, regardless of how physical activity and sedentary time was accumulated over the week [12]. This is somewhat similar to the present study, where the majority of the study population was classified into profiles with relative low/average exercise and relatively high/average sitting time. Moreover, we show a general association between the profiles accumulating more exercise and less sitting time, and lower OR for adverse metabolic health, perceived health, and perceived symptoms. A few studies have used LPA to combine physical activity related variables from different domains for identification of different profiles. For example Hansen et al. used dichotomized data for overall physical activity energy expenditure, active commuting, sitting time at work, and time in moderate-to-vigorous physical activity to identify and characterize patterns of physical activity in 392 participants (30–60 years) [15]. They identified three latent classes: "low-active occupational sitters", "moderately-active commuters", "active energy-spenders". Although not fully comparable to the present profiles, the "low-active occupational sitters"-profile is similar to profile 2 in the present study, and the "active energy-spenders" are similar to profile 6. Comparing these profiles reveal similar characteristic patterns for age and fitness (higher age and lower fitness in the former profiles compared to the latter profiles), but not in sex distribution where the "low-active occupational sitters" were significantly more likely to be women in the Hansen-study with no differences in the present study. However, it should be emphasized that the Hansen-study assessed total physical activity, while the present study assessed exercise.

The present results have several clinical and societal relevant implications. First, the results open up for a different and alternative perspective on how different physical activity-related variables interact within individuals to create combinations that may carry additional information linked to health. For example, identifying and interpreting interactions between more than two variables at the same time can be difficult in traditional variable-centered analyses. In the present study, we identify patterns of naturally occurring interactions of exercise, sitting in leisure time and at work, and cardiorespiratory fitness, pointing to the value of using person-centered analytical approaches such as LPA. The profiles do not only differ in terms of levels (i.e., high vs. low in all variables), they also reveal complex combinations of profile level and shape. This mirrors a substantial heterogeneity of the interaction patterns found and support the importance of using analytical approaches that not only treat heterogeneity as noise/disturbance in models but as valuable information. Second, the main variation in associations between the different profiles and the health outcomes were due to variation in exercise, leisure time sitting, and fitness, which emphasize the need for these three to be targeted for health enhancing interventions. Third, workplace sitting seems to discriminate less for adverse health outcomes, except in those who were "worse off" otherwise (low fitness and low exercise level). This implies that workplace sitting might not be a primary target for health enhancing interventions for all, but for selected sub-groups.

Strengths of this study include the large sample size and the large heterogeneity displayed in the scores on the input variables, demographics, and the perceived health, symptoms, and metabolic measures. Moreover, fitness was estimated using exercise testing, which is considered a strength compared to self-reported fitness. Limitation is the self-report data for exercise, sitting in leisure and at work, as well as the lack of validation of the exercise question. Self-reported data is recognized to induce misclassification bias compared to actual exercise and sitting time, and mixing self-report and objective data for input variable may influence

solutions in the latent profile analysis. Also, the cross-sectional design of the association analyses limits any conclusions of causality and temporal order.

## Conclusions

LPA is particularly suited to capture heterogeneity in a sample and is useful when the aim is to examine complex interaction patterns between several variables and how these interaction patterns relate to various outcomes, which was the aim of the present study. We identified six profiles based on exercise, sitting in leisure time and at work, and cardiorespiratory fitness. There were large variations in the combination of the input variables between profiles, which implies a large variation in exercise, sitting time, and fitness between individuals. Some pairwise similarities were found between profiles (1 and 2, 3 and 4, 5 and 6), mainly based on similar exercise, leisure time sitting, and fitness levels for the pair of profiles. This translated into similar dose-response associations for the pairs with the outcomes. Smaller variations between the paired profiles and associations with the outcomes were explained by differences in sitting at work. In summary, these results emphasize the possibility to target exercise, time spent sitting, and/or fitness in health enhancing promotion intervention and strategies.

## Supporting information

**S1 Table. Fit indices of the estimated latent profile analysis.**
(PDF)

**S2 Table. Drop-out analysis comparing those excluded (n = 19.892) and excluded (n = 45.078) in multi-adjusted analyses in Fig 2.**
(PDF)

## Author Contributions

**Conceptualization:** Elin Ekblom-Bak, Erik Hemmingsson, Lena V. Kallings, Björn Ekblom, Magnus Lindwall.

**Data curation:** Jane Salier Eriksson, Örjan Ekblom.

**Formal analysis:** Elin Ekblom-Bak, Andreas Stenling, Jane Salier Eriksson, Erik Hemmingsson, Örjan Ekblom, Magnus Lindwall.

**Funding acquisition:** Elin Ekblom-Bak.

**Methodology:** Gunnar Andersson, Peter Wallin.

**Project administration:** Gunnar Andersson, Peter Wallin.

**Resources:** Gunnar Andersson, Peter Wallin.

**Writing – original draft:** Elin Ekblom-Bak, Andreas Stenling, Lena V. Kallings, Björn Ekblom, Magnus Lindwall.

**Writing – review & editing:** Elin Ekblom-Bak, Andreas Stenling, Jane Salier Eriksson, Erik Hemmingsson, Lena V. Kallings, Gunnar Andersson, Peter Wallin, Örjan Ekblom, Björn Ekblom, Magnus Lindwall.

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
