## [Decision Letter · Decision Letter 0]

6 Jan 2020

PONE-D-19-27066

Latent profile analysis patterns of physical activity, fitness and sedentary behavior in adults – associations with metabolic risk factors, perceived health, and perceived symptoms

PLOS ONE

Dear Mrs Ekblom-Bak,

Thank you for submitting your manuscript to PLOS ONE. After careful consideration, we feel that it has merit but does not fully meet PLOS ONE’s publication criteria as it currently stands. Therefore, we invite you to submit a revised version of the manuscript that addresses the points raised during the review process.

We would appreciate receiving your revised manuscript by Feb 20 2020 11:59PM. To enhance the reproducibility of your results, we recommend that if applicable you deposit your laboratory protocols in protocols.io, where a protocol can be assigned its own identifier (DOI) such that it can be cited independently in the future. For instructions see: http://journals.plos.org/plosone/s/submission-guidelines#loc-laboratory-protocols

We look forward to receiving your revised manuscript.

Kind regards,

Anne Vuillemin

Academic Editor

PLOS ONE

3. Thank you for stating the following in the Competing Interests: "I have read the journal's policy and the authors of this manuscript have the following competing interests: Gunnar Andersson (responsible for research and method) and Peter Wallin (CEO and responsible for research and method) are employed at HPI Health Profile Institute.".  

We note that one or more of the authors have an affiliation to the commercial funders of this research study : "HPI Health Profile Institute.".

Reviewers' comments:

Reviewer's Responses to Questions

**Comments to the Author**

1. Is the manuscript technically sound, and do the data support the conclusions?

Reviewer #1: Partly

2. Has the statistical analysis been performed appropriately and rigorously? 

Reviewer #1: Yes

3. Have the authors made all data underlying the findings in their manuscript fully available?

Reviewer #1: No

4. Is the manuscript presented in an intelligible fashion and written in standard English?

Reviewer #1: Yes

5. Review Comments to the Author

Reviewer #1: Please see the attached comments.

1. Is the manuscript technically sound, and do the data support the conclusions?

The conclusions were not appropriate.

3. Have the authors made all data underlying the findings in their manuscript fully available?

Not fully available data.

6. PLOS authors have the option to publish the peer review history of their article (what does this mean?). If published, this will include your full peer review and any attached files.

Reviewer #1: No

---

## [Author Response · Author response to Decision Letter 0]

19 Feb 2020

Authors: We are grateful for the many relevant and constructive comments, and we have replied to them point by point below. We sincerely believe that our revision, based on your comments, have improved our manuscript. All changes in the manuscript are highlighted in bold.

Reviewer 1: The purpose of this manuscript was to identify and describe the characteristics of naturally occurring patterns of sedentary time at work and in leisure time, exercise and cardiorespiratory fitness, and the association of such profiles with metabolic risk factors, perceived health and perceived symptoms.

Point 1: Title and Short Title: I suggest that the authors change the titles, replacing the word "physical activity" with "exercise". For in this study the authors evaluate the pattern of exercise and not of physical activity. Although such terms are still commonly used as synonyms, they have different meanings. Every exercise is a physical activity, but not every physical activity is an exercise.

Authors: Thank you for a relevant comment. We agree and have now changed the title accordingly.

Point 2: Conclusions: Lines 47-50: I think that the conclusion can be improved. The conclusion always has to focus on answering the study questions.

Authors: We have now revised the conclusion section of the manuscript

Point 3: Introduction: Lines 54-55: The authors report that “… sustained or decreased levels of exercise [6] …”. This reference refers to the study elaborated by Alan G. Knuth and Pedro C. Hallal entitled: Temporal Trends in Physical Activity: A Systematic Review. This study reported the trend regarding physical activity and not about exercise. Moreover, it seems to me that in the introduction and throughout the manuscript the authors used the terms "physical activity" and "exercise" as synonyms. I suggest that only the term "exercise" be used, as it was this domain that was evaluated in this study. It may be interesting to explain in the introduction the difference between the definitions of these terms: "physical activity" and "exercise".

Authors: Thank you for an important comment. We agree that “exercise” rather than “physical activity” describes what we have evaluated in the study. Accordingly, we have now:

- Included two other references (4 and 6) in the introduction section (page 3) that better highlights the trends in moderate-to-vigorous physical activity (and not overall physical activity), and defined that these references refers to trends in MVPA.

- Defined exercise in the introduction section (page 4).

- Revised throughout the manuscript, so when referring to the results in the study, the term “exercise” is used rather than physical activity. 

Point 4: Introduction: Lines 56-57: Objective studies have identified that exercise represents a small portion of daily physical activity. Therefore, I think the following statement should be changed. “… resulting in greater variability in daily physical activity patterns and physical performance between individuals.”.

Authors: Thank you for this comment. We agree that more recent research implies that exercise represent only a small portion of overall physical activity for the general population. However, there are large intra-individual differences in daily or weekly exercise time (as well as sedentary time), that might, with the generally lower time spent in exercise and higher sedentary time, have been exaggerated the last decades and induced a larger variability in the physical activity pattern. This is what we tried to highlight in this sentence. We have now revised this sentence.

Point 5: Material and methods: I think the authors should inform in the session "Material and methods" the experimental design used in this study.

Authors: We have now stated this in the Material and methods section (page 4).

Point 6: Material and methods: Lines 92-93: The authors report that the participants answered an extensive questionnaire about physical activity habits. How reliable and reproducible is this questionnaire to identify daily physical activity and exercise?

Authors: The exercise question has yet not been validated against objective measures. The sedentary questions were derived from the question previously showed to have high predictive validity, and strong convergent validity with total and prolonged stationary time. We have now added this information in the Material and methods section (page 5).

Point 7: Other variables: Lines 151-57: The authors report that “… Physical activity level prior to the age of 20 years was self-reported by selecting …”. This variable has no reference. It would be interesting for the authors to indicate the reference for the elaboration of this variable. In this variable, is only considered the physical education classes held in the school environment? Another question I have is about the power of the physical education class to represent the level of physical activity prior to 20 years.

Authors: This question has yet not been validated (but work are undertaken to do that). However, in a previously publication, we showed predictive validity for this question regarding exercise level, fitness and health later in life (PMID: 29706117). Regarding what the question consider, the alternatives includes both PE at school and PA outside school hours (leisure time).

Point 8: Results: Line 222: What does “cf.” mean?

Authors: We have made a slight change to this sentence and deleted “cf.” 

Point 9: Results: Lines 285-86: The authors report that “Profile 4 had in general a more beneficial health and symptom profile than profile 3, as did profile 6 compared to profile 5”. In table 2, the numbers show that profile 5 has lower results than profile 6. Therefore, from table 2, profile 5 is better (more beneficial) than 6. Please review this.

Authors: Thank you for this comment. We have now revised this sentence.

Point 10: Discussion: Line 336: The authors report: “physical activity patterns”. Although every exercise is a physical activity, not every physical activity is an exercise. What was subjectively assessed was exercise. So, I think it is more appropriate to use "exercise pattern".

Authors: We have revised this by including “sedentary behaviour, exercise and fitness” instead.

Point 11: Discussion: Lines 334-45: The authors report: “Rather than examining interactions between different types of domains or proxies of physical activity, as we do in the present study …”. I could not find in this paper the analysis of the interactions between the different types of domains or proxies of physical activity, since the present study only assessed exercise.

Authors: We have now deleted this part of the sentence.

Point 12: Discussion: Lines 350-52: The authors report: “Accumulating the recommended amount of weekly physical activity was consistently associated with a more beneficial metabolic health profile.”. Does this statement refer to data from the present study or citation data # 14 or # 29?

Authors: This is referring to the results in reference 31 (that was 29). We have clarified this.

Point 13: Discussion: Lines 352-53: The authors report: “The manner in which this physical activity was accumulated (spread over the week or in just a few days) was largely associated with metabolic risk [29].” I suggest revising this sentence, as the reference cited reports the opposite: “However, the manner in which this activity is accumulated, either spread over most days of the week or compressed into just a couple of days, may have similar associations with the risk factors for the MS. (Am J Health Promot 2010;24[3]:161–169.)”.

Authors: Thank you for pointing this out. This was incorrect reporting of the results, and we have now revised this sentence.

Point 14: Discussion: Line 354: The authors report “where time spent in more physical activity”. I think the authors should state what "more physical activity" means and what intensity of physical activity, because such information is very important.

Authors: We agree that this is important, and have now specified this.

Point 15: Discussion: Lines 355-56: The authors report: “regardless of how physical activity time was accumulated over the week [12].”. This sentence contradicts what is written in the sentence of lines 352-53.

Authors: This part of the sentence is referring to that it did not depend on if the time in MVPA and SED wer accumulated on one day or spread out during the week. We have now added also “sedentary time” to the sentence, as this was shown also for accumulation of sedentary time.

Point 16: Discussion: Lines 358 and 360: I suggest changing the term "physical activity" to "exercise".

Authors: We have now changed this.

Point 17: Discussion: Lines 359-60: The authors report: “… we show a general dose-response in the association between the profiles accumulating more physical activity and less sedentary time …”. I could not find in the text what would be the minimum necessary physical activity, exercise and sedentary time that should be adopted. In addition, I reinforce that the study evaluated exercise and not general physical activity. When considering dose response, we should take into account that the recommendations of physical activity, exercise and more recently sedentary behavior (at least for children and adolescents), the minimum amount of such behaviors for different age groups (children, adolescents, adults and the elderly) vary considerably. Thus, as the present study is composed of adults and the elderly, it might be more interesting to separate these groups.

Authors: Thank you for this comment. We have now deleted the “dose-response” part of this sentence.

Point 18: Discussion: Lines 369-70: As the authors previously reported: Hansen et al. [reference #15] identified and characterized patterns of physical activity. I think we should be very careful to say that: “Comparing these profiles reveal similar characteristic patterns for …”. Since the present study evaluated exercise and not total physical activity. I suggest reviewing the final part of this paragraph.

Authors: We have now added a sentence to emphasize this.

Point 19: Discussion: Lines 392-93: The authors report: “… we identify patterns of naturally occurring interactions of four different domains of physical activity …”. I could not identify in the manuscript the four different domains of physical activity.

Authors: We have now revised this sentence.

Point 20: Discussion: Lines 403-406: Although the authors have presented such limitations for this study, I believe there was a lack of warning about how these limitations may impact the interpretation of the results presented.

Authors: We have now added a sentence regarding this in the limitation section.

Point 21: Discussion: I would like to suggest that the authors address more the clinical relevance of these findings within the discussion.

Authors: Thank you for this comment. We have now elaborated more regarding the implications of the results.

Point 22: Conclusions: Line 413: I suggest changing the term "physical activity" to "exercise".

Authors: We have now changed this.

Point 23: Conclusions: Lines 414-15: The authors report: “and not only overall physical activity level …”. I disagree with the author's statement. For the public health message that must be propagated is that of stimulating the increase of physical activity levels, regardless of the type and intensity. I suggest reading the following papers:

Klenk J, Kerse N. Every step you take. BMJ. 2019;366:l5051. doi: 10.1136/bmj.l5051.

Ekelund U, Tarp J, Steene-Johannessen J, Hansen BH, Jefferis B, Fagerland MW, et al. Dose-response associations between accelerometry measured physical activity and sedentary time and all cause mortality: systematic review and harmonised meta-analysis. BMJ. 2019;366:l4570. doi: 10.1136/bmj.l4570.

Authors: We have now revised the last sentence of the conclusion section.

Point 24: Conclusions: I suggest improving the wording of the conclusion and trying to respond to the objectives.

Authors: We have now revised the conclusion section.

---

## [Decision Letter · Decision Letter 1]

30 Mar 2020

PONE-D-19-27066R1

Latent profile analysis patterns of exercise, sedentary behavior and fitness in adults

– associations with metabolic risk factors, perceived health, and perceived symptoms

PLOS ONE

Dear Mrs Ekblom-Bak,

Thank you for submitting your manuscript to PLOS ONE. After careful consideration, we feel that it has merit but does not fully meet PLOS ONE’s publication criteria as it currently stands. Therefore, we invite you to submit a revised version of the manuscript that addresses the points raised during the review process.

We would appreciate receiving your revised manuscript by May 14 2020 11:59PM. To enhance the reproducibility of your results, we recommend that if applicable you deposit your laboratory protocols in protocols.io, where a protocol can be assigned its own identifier (DOI) such that it can be cited independently in the future. For instructions see: http://journals.plos.org/plosone/s/submission-guidelines#loc-laboratory-protocols

We look forward to receiving your revised manuscript.

Kind regards,

Yoshihiro Fukumoto

Academic Editor

PLOS ONE

Reviewers' comments:

Reviewer's Responses to Questions

**Comments to the Author**

1. If the authors have adequately addressed your comments raised in a previous round of review and you feel that this manuscript is now acceptable for publication, you may indicate that here to bypass the “Comments to the Author” section, enter your conflict of interest statement in the “Confidential to Editor” section, and submit your "Accept" recommendation.

Reviewer #1: All comments have been addressed

Reviewer #2: (No Response)

2. Is the manuscript technically sound, and do the data support the conclusions?

Reviewer #1: Yes

Reviewer #2: Partly

3. Has the statistical analysis been performed appropriately and rigorously? 

Reviewer #1: Yes

Reviewer #2: Yes

4. Have the authors made all data underlying the findings in their manuscript fully available?

Reviewer #1: No

Reviewer #2: Yes

5. Is the manuscript presented in an intelligible fashion and written in standard English?

Reviewer #1: Yes

Reviewer #2: Yes

6. Review Comments to the Author

Reviewer #1: Reviewer 1: The purpose of this manuscript was to identify and describe the characteristics of naturally occurring patterns of sitting time at work and in leisure, exercise and cardiorespiratory fitness, and the association of such profiles with metabolic risk factors, perceived health and perceived symptoms.

Point 1: Title and lines 47, 322, 415:

Sitting time is one of the domains that make up the definition of sedentary behavior. Thus, in order to use a more appropriate definition, since the objective of the study was to assess sitting time, described in the methodology (line 117), I suggest changing the term "sedentary behavior" by "sitting time" in the following places: Title and lines 47, 322, 415.

In addition, I suggest reading these two articles below:

1) Tremblay, M. S., Aubert, S., Barnes, J. D., Saunders, T. J., Carson, V., … Chinapaw, M. J. M. (2017). Sedentary Behavior Research Network (SBRN) – Terminology Consensus Project process and outcome. International Journal of Behavioral Nutrition and Physical Activity, 14(1). doi:10.1186/s12966-017-0525-8

2) Moura, B. P., Rufino, R. L., Faria, R. C., Sasaki, J. E., & Amorim, P. R. S. (2019). Can Replacing Sitting Time with Standing Time Improve Adolescents’ Cardiometabolic Health? International Journal of Environmental Research and Public Health, 16(17), 3115. doi:10.3390/ijerph16173115

The first presents the definitions most used today in the area of sedentary behavior and physical activity research. And the second, presents the benefits of replacing sitting time with standing time.

Point 2: Perceived health and symptoms (lines 143-149) and Other variables (lines 150-161):

The authors report that: This question has yet not been validated (but work are undertaken to do that). However, in a previously publication, we showed predictive validity for this question regarding exercise level, fitness and health later in life (PMID: 29706117). Regarding what the question consider, the alternatives includes both PE at school and PA outside school hours (leisure time).

Therefore, I suggest citing this study in the referred text. This makes it easier for the reader to search for a reference.

Ekblom-Bak, E., Ekblom, Ö., Andersson, G., Wallin, P., & Ekblom, B. (2018). Physical Education and Leisure-Time Physical Activity in Youth Are Both Important for Adulthood Activity, Physical Performance, and Health. Journal of Physical Activity and Health, 15(9), 661–670. doi:10.1123/jpah.2017-0083

Point 3: The PLOS Data policy requires authors to make all data underlying the findings described in their manuscript fully available without restriction, with rare exception (please refer to the Data Availability Statement in the manuscript PDF file). The data should be provided as part of the manuscript or its supporting information, or deposited to a public repository.

Reviewer #2: This study evaluated the pattern of exercise, cardiorespiratory fitness, leisure fitness, and workplace sitting. The authors identified 6 profiles and shows the mean values in systolic blood pressure, diastolic pressure, BMI, symptom, and global health in workers. Then they assessed the association between obesity, hypertension, back pain, stress, global health, or sleeping stress and the profiles. First of all, I appreciate the authors for their meticulous efforts for the revision. However, I have several comments to the authors.

1. The participants are workers so essentially they are healthy enough to work. Can the authors provide the data on hypertension, diabetes, dyslipidemia and metabolic syndrome in the study sample? They also could evaluate the association between such lifestyle disease and the profiles.

2. Even though the authors identified the 6 profiles, the mean values in blood pressure and BMI are within normal ranges. The differences in absolute vales of pain and global health are small. Are these differences meaningful?

3. I think this study message is "these results emphasize the possibility to target exercise, sedentary behaviour, and/or fitness in health enhancing promotion intervention and strategies". I would suggest highlighting this point in abstract and the beginning in the discussion part.

7. PLOS authors have the option to publish the peer review history of their article (what does this mean?). If published, this will include your full peer review and any attached files.

Reviewer #1: No

Reviewer #2: No

---

## [Author Response · Author response to Decision Letter 1]

3 Apr 2020

(Same text below as attached in the file "Response to reviewers"

Review Comments to the Author

Authors: We are grateful for the constructive and relevant comments from the reviewers. We have replied to them point by point below. All changes in the manuscript are highlighted by using “track changes”.

Reviewer #1: Reviewer 1: The purpose of this manuscript was to identify and describe the characteristics of naturally occurring patterns of sitting time at work and in leisure, exercise and cardiorespiratory fitness, and the association of such profiles with metabolic risk factors, perceived health and perceived symptoms.

Point 1: Title and lines 47, 322, 415:

Sitting time is one of the domains that make up the definition of sedentary behavior. Thus, in order to use a more appropriate definition, since the objective of the study was to assess sitting time, described in the methodology (line 117), I suggest changing the term "sedentary behavior" by "sitting time" in the following places: Title and lines 47, 322, 415.

Authors: We agree, and have now made changes in the manuscript where suggested, as well as in lines 31, 120, 177 and 357.

In addition, I suggest reading these two articles below:

1) Tremblay, M. S., Aubert, S., Barnes, J. D., Saunders, T. J., Carson, V., … Chinapaw, M. J. M. (2017). Sedentary Behavior Research Network (SBRN) – Terminology Consensus Project process and outcome. International Journal of Behavioral Nutrition and Physical Activity, 14(1). doi:10.1186/s12966-017-0525-8

2) Moura, B. P., Rufino, R. L., Faria, R. C., Sasaki, J. E., & Amorim, P. R. S. (2019). Can Replacing Sitting Time with Standing Time Improve Adolescents’ Cardiometabolic Health? International Journal of Environmental Research and Public Health, 16(17), 3115. doi:10.3390/ijerph16173115

The first presents the definitions most used today in the area of sedentary behavior and physical activity research. And the second, presents the benefits of replacing sitting time with standing time.

Authors: Thank you for these reading suggestions.

Point 2: Perceived health and symptoms (lines 143-149) and Other variables (lines 150-161):

The authors report that: This question has yet not been validated (but work are undertaken to do that). However, in a previously publication, we showed predictive validity for this question regarding exercise level, fitness and health later in life (PMID: 29706117). Regarding what the question consider, the alternatives includes both PE at school and PA outside school hours (leisure time).

Therefore, I suggest citing this study in the referred text. This makes it easier for the reader to search for a reference.

Ekblom-Bak, E., Ekblom, Ö., Andersson, G., Wallin, P., & Ekblom, B. (2018). Physical Education and Leisure-Time Physical Activity in Youth Are Both Important for Adulthood Activity, Physical Performance, and Health. Journal of Physical Activity and Health, 15(9), 661–670. doi:10.1123/jpah.2017-0083

Authors: Thank you for this comment. We have now added this reference on line 157-158.

Point 3: The PLOS Data policy requires authors to make all data underlying the findings described in their manuscript fully available without restriction, with rare exception (please refer to the Data Availability Statement in the manuscript PDF file). The data should be provided as part of the manuscript or its supporting information, or deposited to a public repository.

Authors: We are aware of that the PLOS Data policy requires the authors to make all data underlying the findings available. However, the data underlying these results are not publicly available because the original approval by the regional ethic's board and the informed consent from the subjects participating in the studies did not include such a direct, free access. However, if a reader wants to access the data, they may contact the holder of the database, HPI Health Profile Institute. We have stated this in the Data Availability Statement (see below).

"The data underlying the findings in our study are not publicly available because the original approval by the regional ethic's board (Stockholm Ethics Review Board, Dnr 2015/1864-31/2 and 2016/9-32) and the informed consent from the subjects participating in the studies did not include such a direct, free access. If a reader wants access to the data underlying the present article, please contact the HPI Health Profile Institute at support@hpihealth.se."

Reviewer #2: This study evaluated the pattern of exercise, cardiorespiratory fitness, leisure fitness, and workplace sitting. The authors identified 6 profiles and shows the mean values in systolic blood pressure, diastolic pressure, BMI, symptom, and global health in workers. Then they assessed the association between obesity, hypertension, back pain, stress, global health, or sleeping stress and the profiles. First of all, I appreciate the authors for their meticulous efforts for the revision. However, I have several comments to the authors.

1. The participants are workers so essentially they are healthy enough to work. Can the authors provide the data on hypertension, diabetes, dyslipidemia and metabolic syndrome in the study sample? They also could evaluate the association between such lifestyle disease and the profiles.

Authors: Thank you for a relevant comment. Unfortunately, we have no such data available for this population. However, we are linking the dataset to national register to be able to perform longitudinal analyses on CVD incidence, hypertension, mortality etc. 

2. Even though the authors identified the 6 profiles, the mean values in blood pressure and BMI are within normal ranges. The differences in absolute vales of pain and global health are small. Are these differences meaningful?

Authors: Thank you for an important comment. To highlight the absolute risk differences between the different profiles (adding to the relative risk differences presented in Figure 2), we are now presenting prevalences of the six dichotomized risk outcomes in relation to the profiles in Table 2.

3. I think this study message is "these results emphasize the possibility to target exercise, sedentary behaviour, and/or fitness in health enhancing promotion intervention and strategies". I would suggest highlighting this point in abstract and the beginning in the discussion part.

Authors: Thank you for this comment, we have bow highlighted this in the abstract and the beginning of the discussion.

---

## [Decision Letter · Decision Letter 2]

10 Apr 2020

Latent profile analysis patterns of exercise, sitting and fitness in adults

– associations with metabolic risk factors, perceived health, and perceived symptoms

PONE-D-19-27066R2

Dear Dr. Ekblom-Bak,

We are pleased to inform you that your manuscript has been judged scientifically suitable for publication and will be formally accepted for publication once it complies with all outstanding technical requirements.

With kind regards,

Yoshihiro Fukumoto

Academic Editor

PLOS ONE

Additional Editor Comments (optional):

Reviewers' comments:

Reviewer's Responses to Questions

**Comments to the Author**

1. If the authors have adequately addressed your comments raised in a previous round of review and you feel that this manuscript is now acceptable for publication, you may indicate that here to bypass the “Comments to the Author” section, enter your conflict of interest statement in the “Confidential to Editor” section, and submit your "Accept" recommendation.

Reviewer #1: All comments have been addressed

Reviewer #2: All comments have been addressed

2. Is the manuscript technically sound, and do the data support the conclusions?

Reviewer #1: Yes

Reviewer #2: Yes

3. Has the statistical analysis been performed appropriately and rigorously? 

Reviewer #1: Yes

Reviewer #2: Yes

4. Have the authors made all data underlying the findings in their manuscript fully available?

Reviewer #1: No

Reviewer #2: No

5. Is the manuscript presented in an intelligible fashion and written in standard English?

Reviewer #1: Yes

Reviewer #2: Yes

6. Review Comments to the Author

Reviewer #1: In question 4 I marked "No", as there are only two possible answers (yes or no). But the authors reported the reason for not making the data publicly available.

Reviewer #2: (No Response)

7. PLOS authors have the option to publish the peer review history of their article (what does this mean?). If published, this will include your full peer review and any attached files.

Reviewer #1: No

Reviewer #2: Yes: Kotaro Nochioka

---

## [Editor Report · Acceptance letter]

14 Apr 2020

PONE-D-19-27066R2 

Latent profile analysis patterns of exercise, sitting and fitness in adults
– associations with metabolic risk factors, perceived health, and perceived symptoms 

Dear Dr. Ekblom-Bak:

I am pleased to inform you that your manuscript has been deemed suitable for publication in PLOS ONE. Congratulations! Your manuscript is now with our production department. 

With kind regards,

on behalf of

Dr. Yoshihiro Fukumoto 

Academic Editor

PLOS ONE